# Achieving $\mathcal{O}(\epsilon^{-1.5})$ Complexity in Hessian/Jacobian-free Stochastic Bilevel Optimization

**Yifan Yang, Peiyao Xiao and Kaiyi Ji**
Department of Computer Science and Engineering
University at Buffalo
Buffalo, NY 14260
{yyang99, peiyaoxi, kaiyiji}@buffalo.edu

## Abstract

In this paper, we revisit the bilevel optimization problem, in which the upper-level objective function is generally nonconvex and the lower-level objective function is strongly convex. Although this type of problem has been studied extensively, it still remains an open question how to achieve an $\mathcal{O}(\epsilon^{-1.5})$ sample complexity in Hessian/Jacobian-free stochastic bilevel optimization without any second-order derivative computation. To fill this gap, we propose a novel Hessian/Jacobian-free bilevel optimizer named FdeHBO, which features a simple fully single-loop structure, a projection-aided finite-difference Hessian/Jacobian-vector approximation, and momentum-based updates. Theoretically, we show that FdeHBO requires $\mathcal{O}(\epsilon^{-1.5})$ iterations (each using $\mathcal{O}(1)$ samples and only first-order gradient information) to find an $\epsilon$-accurate stationary point. As far as we know, this is the first Hessian/Jacobian-free method with an $\mathcal{O}(\epsilon^{-1.5})$ sample complexity for nonconvex-strongly-convex stochastic bilevel optimization.

## 1 Introduction

Bilevel optimization has drawn intensive attention due to its wide applications in meta-learning [18, 4, 50], hyperparameter optimization [18, 52, 14], reinforcement learning [35, 27], signal process [36, 16] and communication [31] and federated learning [59]. In this paper, we study the following stochastic bilevel optimization problem.

$$\min_{x \in \mathbb{R}^p} \Phi(x) = f(x, y^*(x)) := \mathbb{E}_\xi \left[ f(x, y^*(x); \xi) \right]$$
$$\text{s.t. } y^*(x) = \arg\min_{y \in \mathbb{R}^q} g(x, y) := \mathbb{E}_\zeta \left[ g(x, y^*(x); \zeta) \right] \tag{1}$$

where the upper- and lower-level objective functions $f(x, y)$ and $g(x, y)$ take the expectation form w.r.t. the random variables $\xi$ and $\zeta$, and are jointly continuously differentiable. In this paper, we focus on the nonconvex-strongly-convex bilevel setting, where the lower-level function $g(x, \cdot)$ is strongly convex and the upper-level function $\Phi(x)$ is nonconvex. This class of bilevel problems has been studied extensively from the theoretical perspective in recent years. Among them, [19, 30, 3, 62] proposed bilevel approaches with a double-loop structure, which update $x$ and $y$ in a nested manner. Single-loop bilevel algorithms have also attracted significant attention recently [27, 62, 34, 25, 9, 40, 11] due to the simple updates on all variables simultaneously. Among them, the approaches in [62, 34, 25] have been shown to achieve an $\mathcal{O}(\epsilon^{-1.5})$ sample complexity, but with expensive evaluations of Hessian/Jacobian matrices or Hessian/Jacobian-vector products.

Hessian/Jacobian-free bilevel optimization has received increasing attention due to its high efficiency and feasibility in practical large-scale settings. In particular, [15, 48, 61] directly ignored the

37th Conference on Neural Information Processing Systems (NeurIPS 2023).

| Algorithm | Samples | Batch size | # of iterations | Loops per iteration |
|-----------|---------|------------|-----------------|---------------------|
| PZOBO-S [58] | $\widetilde{\mathcal{O}}(p^2\epsilon^{-3})$ | $\mathcal{O}(\epsilon^{-1})$ | $\widetilde{\mathcal{O}}(p^2\epsilon^{-2})$ | 2 |
| F$^2$SA [37] | $\widetilde{\mathcal{O}}(\epsilon^{-3.5})$ | $\mathcal{O}(1)$ | $\widetilde{\mathcal{O}}(\epsilon^{-3.5})$ | 1 |
| F$^3$SA [37] | $\widetilde{\mathcal{O}}(\epsilon^{-2.5})$ | $\mathcal{O}(1)$ | $\widetilde{\mathcal{O}}(\epsilon^{-2.5})$ | 1 |
| FdeHBO (this paper) | $\widetilde{\mathcal{O}}(\epsilon^{-1.5})$ | $\mathcal{O}(1)$ | $\widetilde{\mathcal{O}}(\epsilon^{-1.5})$ | 1 |

Table 1: Comparison of stochastic Hessian/Jacobian-free bilevel optimization algorithms.

computation of all second-order derivatives. However, such eliminations may lead to performance degeneration [2, 13], and can vanish the hypergradient for bilevel problems with single-variable upper-level function, i.e., $\Phi(x) = f(y^*(x))$. [56, 23] proposed zeroth-order approaches that approximate the hypergradient using only function values. These methods do not have a convergence rate guarantee. Recently, several Hessian/Jacobian-free bilevel algorithms were proposed by [42, 57, 53, 8] by reformulating the lower-level problem into the optimality-based constraints such as $g(x,y) \leq \min_y g(x,y)$. However, these approaches all focus on the deterministic setting, and their extensions to the stochastic setting remain unclear. In the stochastic case, [58] proposed evolution strategies based bilevel method, which achieves a high sample complexity of $\mathcal{O}(p^2\epsilon^{-2})$, where $p$ is the problem dimension. Most recently, [37] proposed two fully first-order (i.e., Hessian/Jacobian-free) value-function-based stochastic bilevel optimizer named F$^2$SA and its momentum-based version F$^3$SA with a single-loop structure, which achieves sample complexities of $\mathcal{O}(\epsilon^{-3.5})$ and $\mathcal{O}(\epsilon^{-2.5})$, respectively. However, there is still a large gap of $\epsilon^{-1}$, compared to the optimal complexity of $\mathcal{O}(\epsilon^{-1.5})$. Then, an important open question, as recently proposed by [37], is:

- Can we achieve an $\mathcal{O}(\epsilon^{-1.5})$ sample/gradient complexity for nonconvex-strongly-convex bilevel optimization using only first-order gradient information?

## 1.1 Our Contributions

In this paper, we provide an affirmative answer to the above question by proposing a new Hessian/Jacobian-free stochastic bilevel optimizer named FdeHBO with three main features. First, FdeHBO takes the fully single-loop structure with momentum-based updates on three variables $y, v$ and $x$ for optimizing the lower-level objective, the linear system (LS) of the Hessian-inverse-vector approximation, and the upper-level objective, respectively. Second, FdeHBO contains only a single matrix-vector product at each iteration, which admits a simple first-order finite-difference estimation. Third, FdeHBO involves an auxiliary projection on $v$ updates to ensure the boundedness of the Hessian-vector approximation error, the variance on momentum-based iterates, and the smoothness of the LS loss function. Our detailed contributions are summarized below.

- Theoretically, we show that FdeHBO achieves a sample/gradient complexity of $\mathcal{O}(\epsilon^{-1.5})$ and an iteration complexity of $\mathcal{O}(\epsilon^{-1.5})$ to achieve an $\epsilon$-accurate stationary point, both of which outperforms existing results by a large margin. As far as we know, this is the first-known method with an $\mathcal{O}(\epsilon^{-1.5})$ sample complexity for nonconvex-strongly-convex stochastic bilevel optimization using only first-order gradient information.

- Technically, we show that the auxiliary projection can provide more accurate iterates on $v$ in solving the LS problem without affecting the overall convergence behavior, and in addition, provide a novel characterization of the gradient estimation error and the iterative progress during the $v$ updates, as well as the impact of the $y$ and $v$ updates on the momentum-based hypergradient estimation, all of which do not exist in previous studies. In addition, the finite-different approximations make the unbiased assumptions in the momentum-based gradients no longer hold, and hence a more careful analysis is required.

- As a byproduct, we further propose a fully single-loop momentum-based method named FMBO in the small-dimensional case with matrix-vector-based hypergradient computations. Differently from existing momentum-based bilevel methods with $\mathcal{O}(\log \frac{1}{\epsilon})$ Hessian-vector evaluations per iteration, FMBO contains only a single Hessian-vector computation per iteration with the same $\mathcal{O}(\epsilon^{-1.5})$ sample complexity.

We also want to emphasize our technical differences from previous works as below.

**Comparison to existing momentum-based methods.** Previous momentum-based methods [62, 34] solve the linear system (LS) to a high accuracy of $\mathcal{O}(\epsilon)$, whereas our algorithm includes a new estimation error by the single-step momentum update on LS, and this error is also correlated with the lower-level updating error and the hypergradient estimation error. In addition, due to the finite-difference approximation, the stochastic gradients in all three updates on $y, v, x$ are no longer unbiased. Non-trivial efforts need to be taken to deal with such challenges and derive the optimal complexity.

**Comparison to existing fully single-loop methods.** The analysis of the single-step momentum update in solving the LS requires the smoothness of the LS loss function and the boundedness of LS gradient variance, both of which may not be satisfied. To this end, we include an auxiliary projection and show it not only guarantees these crucial properties, but also, in theory, provides an improved per-iteration progress. As a comparison, existing works on fully single-loop stochastic bilevel optimization such as SOBA/SABA [11] and FLSA [40] with a new time scale to update the LS problem often assume that the iterates on $v$ are bounded during the process. We do not require such assumptions. In addition, an $\mathcal{O}(\epsilon^{-1.5})$ complexity has not been established for fully single-loop bilevel algorithms yet.

## 1.2 Related Work

**Bilevel optimization methods.** Bilevel optimization, which was first introduced by [6], has been studied for decades. By replacing the lower-level problem with its optimality conditions, [26, 20, 54, 55] reformulated the bilevel problem to the single-level problem. Gradient-based bilevel methods have shown great promise recently, which can be divided into approximate implicit differentiation (AID) [12, 49, 41, 3] and iterative differentiation (ITD) [47, 17, 15, 52, 21] based approaches. Recently, a bunch of stochastic bilevel algorithms has been proposed via Neumann series [9, 30], recursive momentum [62, 28, 25] and variance reduction [62, 11]. Theoretically, the convergence of bilevel optimization has been analyzed by [18, 52, 45, 19, 30, 27, 3, 11]. Among them, [29] provides the lower complexity bounds for deterministic bilevel optimization with (strongly-)convex upper-level functions. [25, 9, 62, 34] achieved the near-optimal sample complexity with second-order derivative computations. Some works studied deterministic bilevel optimization with convex or Polyak-Lojasiewicz (PL) lower-level problems via mixed gradient aggregation [51, 46, 39], log-barrier regularization [45], primal-dual method [57] and dynamic barrier [63]. More results and details can be found in the survey by [44].

**Hessian/Jacobian-free bilevel optimization.** Some Hessian/Jacobian-free bilevel optimization methods have been proposed recently by [58, 43, 15, 23, 56, 48]. Among them, FOMAML [15, 48] and MUMOMAML [61] directly ignore the computation of all second-order derivatives. Several Hessian/Jacobian-free bilevel algorithms were proposed by [42, 57, 53, 8] by replacing the lower-level problem with the optimality conditions as the constraints. However, these approaches focus only on the deterministic setting. Recently, zeroth-order stochastic approaches have been proposed for the hypergradient estimation [56, 23, 58]. Theoretically, [58] analyzed the convergence rate for their method. [37] proposed fully first-order stochastic bilevel optimization algorithms based on the value-function-based lower-level problem reformulation. This paper proposes a new Hessian/Jacobian-free stochastic bilevel algorithm that for the first time achieves an $\mathcal{O}(\epsilon^{-1.5})$ sample complexity.

**Momentum-based bilevel approaches.** The recursive momentum technique was first introduced by [10, 60] for minimization problems to improve the SGD-based updates in theory and in practice. This technique has been incorporated in stochastic bilevel optimization [34, 9, 24, 25, 62]. These approaches involve either Hessian-inverse matrix computations or a subloop of a number of iterations in the Hessian-inverse-vector approximation. As a comparison, our proposed method takes the simpler fully single-loop structure, and only uses the first-order gradient information.

**Finite-difference matrix-vector approximation.** The finite-difference matrix-vector estimation has been studied extensively in the problems of escaping from saddle points [1] [7] (some other related works can be found therein), neural architecture search (NAS) [43] and meta-learning [13]. However, such finite-different estimation can be sensitive to the selection of the smoothing constant, and may suffer from some numerical issues in practice [32][33], such as rounding errors. It is interesting but still open to developing a fully first-order stochastic bilevel optimizer without the finite-different matrix-vector estimation. We would like to leave it for future study.

**Algorithm 1** Hessian/Jacobian-free Bilevel Optimizer via Projection-aided Finite-difference Estimation

1: **Input:** $\{\alpha_t, \beta_t, \lambda_t\}_{t=0}^{T-1}$ and $r_v$.
2: **Initialize:**
3: **for** $t = 0, 1, 2, ..., T-1$ **do**
4:     Compute the gradient estimator $h_t^g$ by eq. (6) and update $y_{t+1} = y_t - \beta_t h_t^g$.
5:     Compute the gradient estimator $h_t^R$ by eq. (7) and update $w_{t+1} = v_t - \lambda_t \widetilde{h}_t^R$.
6:     Set $v_{t+1} = \begin{cases} w_{t+1}, & \|w_{t+1}\| \le r_v; \\ \frac{r_v w_{t+1}}{\|w_{t+1}\|}, & \|w_{t+1}\| > r_v. \end{cases}$
7:     Compute the gradient estimator $h_t^f$ by eq. (10) and update $x_{t+1} = x_t - \alpha_t \widetilde{h}_t^f$.
8: **end for**

## 2 Algorithms

In this section, we first describe the hypergradient computation in bilevel optimization, and then present the proposed Hessian/Jacobian-free bilevel method.

### 2.1 Hypergradient Computation

One major challenge in bilevel optimization lies in computing the hypergradient $\nabla\Phi(x)$ due to the implicit and complex dependence of the lower-level minimizer $y^*$ on $x$. To see this, if $g$ is twice differentiable, $\nabla_y g$ is continuously differentiable and the Hessian $\nabla_{yy}^2 g(x, y^*(x))$ is invertible, using the implicit function theorem (IFT) [22, 5], the hypergradient $\nabla\Phi(x)$ takes the form of

$$\nabla\Phi(x) = \nabla_x f(x, y^*) - \nabla_{xy}^2 g(x, y^*) \big[\nabla_{yy}^2 g(x, y^*)\big]^{-1} \nabla_y f(x, y^*). \tag{2}$$

Note that the hypergradient in eq. (2) requires computing the exact solution $y^*$ and the expensive Hessian inverse $[\nabla_{yy}^2 g(x, y^*)]^{-1}$. To approximate this hypergradient efficiently, we define the following (stochastic) hypergradient surrogates as

$$\begin{aligned} \bar{\nabla} f(x, y, v) &= \nabla_x f(x, y) - \nabla_{xy}^2 g(x, y) v, \\ \bar{\nabla} f(x, y, v; \xi) &= \nabla_x f(x, y; \xi) - \nabla_{xy}^2 g(x, y; \xi) v, \end{aligned} \tag{3}$$

where $v \in \mathbb{R}^q$ is an auxiliary vector to approximate the Hessian-inverse-vector product in eq. (2), and $\bar{\nabla} f(x, y, v; \xi)$ can be regarded as a stochastic version of $\bar{\nabla} f(x, y, v)$. Based on eq. (3), one needs to find an efficient estimate $y$ of $y^*$, e.g., via an iterative optimization procedure, as well as a feasible estimate $v$ of the solution $v^* = [\nabla_{yy}^2 g(x, y)]^{-1} \nabla_y f(x, y)$ of a linear system (LS) (equivalently quadratic programming) whose generic loss function is given by

$$\text{(Linear system loss:)} \quad R(x, y, v) = \frac{1}{2} v^T \nabla_{yy}^2 g(x, y) v - v^T \nabla_y f(x, y), \tag{4}$$

where the gradient of $R(x, y, v)$ w.r.t. $v$ is given by

$$\nabla_v R(x, y, v) = \nabla_{yy}^2 g(x, y) v - \nabla_y f(x, y). \tag{5}$$

Similarly to eq. (3), we also define $\nabla_v R(x, y, v; \psi) = \nabla_{yy}^2 g(x, y; \psi) v - \nabla_y f(x, y; \psi)$ over any sample $\psi$ as a stochastic version of $\nabla_v R(x, y, v)$ in eq. (5). It can be seen from eq. (3), eq. (4) and eq. (5) that the updates on the LS system involve the Hessian- and Jacobian-vector products, which can be computationally intractable in the high-dimensional case. In the next section, we propose a novel stochastic Hessian/Jacobian-free bilevel algorithm.

### 2.2 Hessian/Jacobian-free Bilevel Optimizer via Projection-aided Finite-difference Estimation

As shown in Algorithm 1, we propose a fully single-loop stochastic Hessian/Jacobian-free bilevel optimizer named FdeHBO via projection-aided finite-difference estimation. It can be seen that FdeHBO first minimizes the lower-level objective function $g(x, y)$ w.r.t. $y$ by running a single-step momentum-based update as $y_{t+1} = y_t - \beta_t h_t^g$, where $\beta_t$ is the stepsize and $h_t^g$ is the momentum-based gradient estimator that takes the form of

$$h_t^g = \eta_t^g \nabla_y g(x_t, y_t; \zeta_t) + (1 - \eta_t^g)\big(h_{t-1}^g + \nabla_y g(x_t, y_t; \zeta_t) - \nabla_y g(x_{t-1}, y_{t-1}; \zeta_t)\big) \tag{6}$$

where $\eta_t^g \in [0, 1]$ is a tuning parameter. The next key step is to deal with the LS problem via solving the quadratic problem eq. (4) as $w_{t+1} = v_t - \lambda_t \tilde{h}_t^R$, with the momentum-based gradient $\tilde{h}_t^R$ given by

$$\tilde{h}_t^R = \eta_t^R \widetilde{\nabla}_v R(x_t, y_t, v_t, \delta_\epsilon; \psi_t) + (1 - \eta_t^R)\big(h_{t-1}^R + \widetilde{\nabla}_v R(x_t, y_t, v_t, \delta_\epsilon; \psi_t)$$
$$- \widetilde{\nabla}_v R(x_{t-1}, y_{t-1}, v_{t-1}, \delta_\epsilon; \psi_t)\big), \tag{7}$$

where $\widetilde{\nabla}_v R$ is a Hessian-free version of the LS gradient $\nabla_v R$ in eq. (5), given by

(First-order LS gradient:)  $\widetilde{\nabla}_v R(x_t, y_t, v_t, \delta_\epsilon; \psi_t) = \widetilde{H}(x_t, y_t, v_t, \delta_\epsilon; \psi_t) - \nabla_y f(x_t, y_t; \psi_t)$. (8)

Note that in the above eq. (8), $\widetilde{H}(x_t, y_t, v_t, \delta_\epsilon; \psi_t)$ is the finite-difference estimation of the Hessian-vector product $\nabla_{yy}^2 g(x_t, y_t; \psi_t)v_t$, which takes the form of

$$\widetilde{H}(x_t, y_t, v_t, \delta_\epsilon; \psi_t) = \frac{\nabla_y g(x_t, y_t + \delta_\epsilon v_t; \psi_t) - \nabla_y g(x_t, y_t - \delta_\epsilon v_t; \psi_t)}{2\delta_\epsilon}, \tag{9}$$

where $\delta_\epsilon > 0$ is a small constant. Note that in eq. (9), if the iterative $v_t$ is unbounded, the approximation error between $\widetilde{H}$ and $\nabla_{yy}^2 g(x_t, y_t; \psi_t)v_t$ can be uncontrollable as well. We further prove lemma 5 in appendix B that the bound of this gap relies on $\|v_t\|$ and $\delta$ but it is independent of the dimension of $y_t$. To this end, after obtaining $w_{t+1}$, our key step in line 6 introduces an auxiliary projection on a ball (which can be generalized to any convex and bounded domain) with a radius of $r_v$ as

$$\text{(Auxiliary projection)} \quad v_{t+1} = \begin{cases} w_{t+1}, & \|w_{t+1}\| \le r_v; \\ \frac{r_v w_{t+1}}{\|w_{t+1}\|}, & \|w_{t+1}\| > r_v. \end{cases}$$

This auxiliary projection guarantees the boundedness of $v_t, t = 0, ..., T - 1$, which serves **three** important purposes. First, it ensures the smoothness of the LS loss function $R(x, y, v)$ in eq. (5) w.r.t. all $x, y$ and $v$, which is crucial in the convergence analysis of the momentum-based updates. Second, the boundedness of $v_t$ also ensures that the estimation variance of the stochastic LS gradient $\nabla_v R(x_t, y_t, v_t; \psi_t)$ does not explode. Third, it guarantees the error of the finite-difference Hessian-vector approximation to be sufficiently small with proper $\delta_\epsilon$. We will show later that under a proper choice of the radius $r_v$, this auxiliary projection provides better per-step progress, and the proposed algorithm achieves a stronger convergence performance. Finally, for the upper-level problem, the momentum-based hypergradient estimate $\tilde{h}_t^f$ is designed as

$$\tilde{h}_t^f = \eta_t^f \widetilde{\nabla} f(x_t, y_t, v_t, \delta_\epsilon; \bar{\xi}_t) + (1 - \eta_t^f)\big(h_{t-1}^f + \widetilde{\nabla} f(x_t, y_t, v_t, \delta_\epsilon; \bar{\xi}_t)$$
$$- \widetilde{\nabla} f(x_{t-1}, y_{t-1}, v_{t-1}, \delta_\epsilon; \bar{\xi}_t)\big), \tag{10}$$

where $\widetilde{\nabla} f(x, y, v, \delta_\epsilon; \bar{\xi}_t)$ is the fully first-order hypergradient estimate evaluated at two consecutive iterates $(x_t, y_t, v_t)$ and $(x_{t-1}, y_{t-1}, v_{t-1})$ is given by

$$\widetilde{\nabla} f(x, y, v, \delta_\epsilon; \bar{\xi}_t) = \nabla_x f(x, y; \bar{\xi}_t) - \widetilde{J}(x, y, v, \delta_\epsilon; \bar{\xi}_t),$$

and $\widetilde{J}(x, y, v, \delta_\epsilon; \bar{\xi}_t)$ is the finite-difference Jacobian-vector approximation given by

$$\widetilde{J}(x, y, v, \delta_\epsilon; \bar{\xi}_t) := \frac{\nabla_x g(x, y + \delta_\epsilon v; \bar{\xi}_t) - \nabla_x g(x, y - \delta_\epsilon v; \bar{\xi}_t)}{2\delta_\epsilon}. \tag{11}$$

Note that $\widetilde{\nabla}_v R$ and $\widetilde{\nabla} f$ are **biased** estimators of the gradients $\nabla_v R$ and $\bar{\nabla} f$, which further complicates the convergence analysis on the momentum-based updates because the conventional analysis on the recursive momentum requires the unbiased gradient estimation to ensure the variance reduction effect. By controlling the perturbation $\delta_\epsilon$ properly, we will show that FdeHBO can achieve an $\mathcal{O}(\epsilon^{-1.5})$ convergence and complexity performance without any second-order derivative computation.

## 2.3 Extension to Small-Dimensional Case

As a byproduct of our proposed FdeHBO, we further propose a fully single-loop momentum-based bilevel optimizer (FMBO), which is more suitable in the small-dimensional case without finite-difference approximation. As shown in Algorithm 2, FMBO first takes the same lower-level updates

---

**Algorithm 2** Fully Single-loop Momentum-based Bilevel Optimizer (FMBO)

---

1: **Input:** $\{\alpha_t, \beta_t, \lambda_t\}_{t=0}^{T-1}$, and $r_v$.
2: **Initialize:**
3: **for** $t = 0, 1, 2, ..., T-1$ **do**
4:     Compute the gradient estimator $h_t^g$ by eq. (6) and update $y_{t+1} = y_t - \beta_t h_t^g$.
5:     Compute the gradient estimator $h_t^R$ by eq. (12) and update $w_{t+1} = v_t - \lambda_t h_t^R$.
6:     Set $v_{t+1} = \begin{cases} w_{t+1}, & \|w_{t+1}\| \leq r_v; \\ \frac{r_v w_{t+1}}{\|w_{t+1}\|}, & \|w_{t+1}\| > r_v. \end{cases}$
7:     Compute the gradient estimator $h_t^f$ by eq. (13) and update $x_{t+1} = x_t - \alpha_t h_t^f$.
8: **end for**

---

on $y_t$ as in eq. (6). Then, it solves the LS problem as $w_{t+1} = v_t - \lambda_t h_t^R$, where the momentum-based gradient estimator is given by

$$
\begin{aligned}
h_t^R =& \eta_t^R \nabla_v R(x_t, y_t, v_t; \psi_t) + (1 - \eta_t^R)\big(h_{t-1}^g + \nabla_v R(x_t, y_t, v_t; \psi_t) \\
& - \nabla_v R(x_{t-1}, y_{t-1}, v_{t-1}; \psi_t)\big),
\end{aligned} \tag{12}
$$

where differently from FdeHBO, we here use the precise gradient $\nabla_v R$ without finite-difference approximation. Similarly to FdeHBO, we add an auxiliary projection on the $v_t$ updates to ensure the LS smoothness and bounded variance. Finally, for the upper-level problem, we optimize $x_t$ based on a momentum-based update as $x_{t+1} = x_t - \alpha_t h_t^f$ with the hypergradient estimator

$$
h_t^f = \eta_t^f \bar{\nabla} f(x_t, y_t, v_t; \bar{\xi}_t) + (1 - \eta_t^f)(h_{t-1}^f + \bar{\nabla} f(x_t, y_t, v_t; \bar{\xi}_t) - \bar{\nabla} f(x_{t-1}, y_{t-1}, v_{t-1}; \bar{\xi}_t)) \tag{13}
$$

where $\eta_t^f \in [0, 1]$ is a tuning parameter. Similarly, we directly use the hypergradient estimate in eq. (3) without the finite-difference estimation. We note that compared to existing momentum-based algorithms [62, 34] that contains $\mathcal{O}(\log \frac{1}{\epsilon})$ steps in solving the LS problem, FMBO takes the fully single-loop structure with a single-step momentum-based acceleration on the LS updates.

## 3 Main Results

### 3.1 Assumptions and Definitions

We make the following standard assumptions for the upper- and lower-level objective functions, as also adopted by [30, 9, 34]. The following assumption imposes the Lipschitz condition on the upper-level function $f(x, y)$.

**Assumption 1.** *For any $x \in \mathbb{R}^{d_x}$ and $y \in \mathbb{R}^{d_y}$, there exist positive constants $L_{f_x}$, $L_{f_y}$, $C_{f_x}$ and $C_{f_y}$ such that $\nabla_x f(x, y)$ and $\nabla_y f(x, y)$ are $L_{f_x}$- and $L_{f_y}$-Lipschitz continuous w.r.t. $(x, y)$, and $\|\nabla_x f(x, y)\|^2 \leq C_{f_x}$, $\|\nabla_y f(x, y)\|^2 \leq C_{f_y}$.*

The following assumption imposes the Lipschitz condition on the lower-level function $g(x, y)$.

**Assumption 2.** *For any $x \in \mathbb{R}^{d_x}$ and $y \in \mathbb{R}^{d_y}$, there exist positive constants $\mu_g$, $L_g$, $L_{g_{xy}}$, $L_{g_{yy}}$, $C_{g_{xy}}, C_{g_{yy}}$ such that*

- *Function $g(x, y)$ is twice continuously differentiable;*

- *Function $g(x, \cdot)$ is $\mu_g$-strongly-convex;*

- *The derivatives $\nabla_y g(x, y)$, $\nabla_{xy}^2 g(x, y)$ and $\nabla_{yy}^2 g(x, y)$ are $L_g$-, $L_{g_{xy}}$- and $L_{g_{yy}}$-Lipschitz continuous w.r.t. $(x, y)$;*

- *$\|\nabla_{xy}^2 g(x, y)\|^2 \leq C_{g_{xy}}$ and $\|\nabla_{yy}^2 g(x, y)\|^2 \leq C_{g_{yy}}$.*

The following assumption is adopted for the stochastic functions $f(x, y; \xi)$ and $g(x, y; \zeta)$.

**Assumption 3.** *Assumptions 1 and 2 hold for $f(x, y; \xi)$ and $g(x, y; \zeta)$ for $\forall \xi$ and $\zeta$. Moreover, we assume that there exist positive constants $\sigma_{f_x}$, $\sigma_{f_y}$, $\sigma_g$, $\sigma_{g_{xy}}$ and $\sigma_{g_{yy}}$ such that*

$$\mathbb{E}\left[\|\nabla_x f(x, y) - \nabla_x f(x, y; \xi)\|^2\right] \leq \sigma_{f_x}^2, \quad \mathbb{E}\left[\|\nabla_y f(x, y) - \nabla_y f(x, y; \xi)\|^2\right] \leq \sigma_{f_y}^2,$$

$$\mathbb{E}\left[\|\nabla_y g(x, y) - \nabla_y g(x, y; \zeta)\|^2\right] \leq \sigma_g^2, \quad \mathbb{E}\left[\|\nabla_{xy}^2 g(x, y) - \nabla_{xy}^2 g(x, y; \xi)\|^2\right] \leq \sigma_{g_{xy}}^2,$$

$$\mathbb{E}\left[\|\nabla_{yy}^2 g(x, y) - \nabla_{yy}^2 g(x, y; \xi)\|^2\right] \leq \sigma_{g_{yy}}^2.$$

**Definition 1.** *We say $\bar{x}$ is an $\epsilon$-accurate stationary point of a function $\Phi(x)$ if $\mathbb{E}\|\nabla\Phi(\bar{x})\|^2 \leq \epsilon$, where $\bar{x}$ is the output of an optimization algorithm.*

### 3.2 Convergence and Complexity Analysis of FdeHBO

We further provide the convergence analysis for the proposed Hessian/Jacobian-free FdeHBO algorithm. We first characterize several estimation properties of FdeHBO. Let $e_t^f := \widetilde{h}_t^f - \nabla f(x_t, y_t, v_t) - \Delta(x_t, y_t, v_t)$ denote the hypergradient estimation error.

**Proposition 1.** *Under Assumption 3, the iterates of the outer problem by Algorithm 1 satisfy*

$$\mathbb{E}\|e_{t+1}^f\|^2 \leq \left[(1 - \eta_{t+1}^f)^2 + 4L_{g_{xy}}r_v^2\delta_\epsilon\right]\mathbb{E}\|e_t^f\|^2 + 4(\eta_{t+1}^f)^2\sigma_f^2 + \left(4L_{g_{xy}}r_v^4\delta_\epsilon + 16L_{g_{xy}}^2 r_v^4\delta_\epsilon^2\right)$$

$$+ 6(1 - \eta_{t+1}^f)^2\left[L_F^2\alpha_t^2\mathbb{E}\|\widetilde{h}_t^f\|^2 + 2L_F^2\beta_t^2\left(\mathbb{E}\|e_t^g\|^2 + \|\nabla_y g(x_t, y_t)\|^2\right)\right.$$

$$\left. + 2C_{g_{xy}}\lambda_t^2\left(\mathbb{E}\|e_t^R\|^2 + L_g^2\mathbb{E}\|v_t - v_t^*\|^2\right)\right],$$

*for all $t \in \{0, ..., T-1\}$ with $L_F^2 = 2\left(L_{f_x}^2 + L_{g_{xy}}^2 r_v^2\right)$.*

The hypergradient estimator error $\mathcal{O}(\mathbb{E}\|e_{t+1}^f\|^2)$ contains three main components. The first term $[(1 - \eta_{t+1}^f)^2 + 4L_{g_{xy}}r_v^2\delta_\epsilon]\mathbb{E}\|e_t^f\|^2$ indicates the per-iteration improvement induced by the momentum-based update, the error term $\alpha_t^2\mathbb{E}\|h_t^f\|^2$ is caused by the $x_t$ updates, the error term $\mathcal{O}(\beta_t^2\mathbb{E}(\|e_t^g\|^2 + \|\nabla_y g(x_t, y_t)\|^2))$ is caused by solving the lower-level problem, and the new error term $\mathcal{O}(\lambda_t^2\mathbb{E}(\|e_t^R\|^2 + L_g^2\|v_t - v_t^*\|^2))$ is induced by the one-step momentum update on the LS problem, which does not exist in previous momentum-based bilevel methods [62, 34, 25] that solve the LS problem to a high accuracy. Also note that the errors $4L_{g_{xy}}r_v^2\delta_\epsilon\mathbb{E}\|e_t^f\|^2$ and $4L_{g_{xy}}r_v^2\delta_\epsilon + 16L_{g_{xy}}^2 r_v^4\delta_\epsilon^2$ are caused by the finite-difference approximation error. Fortunately, by choosing the perturbation level $\delta_\epsilon$ in these two terms to be properly small, it can guarantee the descent factor $(1 - \eta_{t+1}^f)^2 + 4L_{g_{xy}}r_v^2\delta_\epsilon$ to be at an order of $(1 - \mathcal{O}(\eta_{t+1}^f))^2$, and hence the momentum-based variance reduction effect is still applied.

**Proposition 2.** *For $\forall \psi$, define $e_t^R := \widetilde{h}_t^R - \nabla_v R(x_t, y_t, v_t)$. Under Assumptions 1, 2, 3, we have*

$$\mathbb{E}\|e_{t+1}^R\|^2 \leq \left[(1 - \eta_{t+1}^R)^2(1 + 96L_g^4\lambda_t^2) + 4L_{g_{yy}}r_v^2\delta_\epsilon\right]\mathbb{E}\|e_t^R\|^2 + \left(4L_{g_{yy}}r_v^2\delta_\epsilon + 8L_{g_{yy}}^2 r_v^4\delta_\epsilon^2\right)$$

$$+ 8(\eta_{t+1}^R)^2(\sigma_{g_{yy}}^2 r_v^2 + \sigma_{f_y}^2) + 96(1 - \eta_{t+1}^R)^2 L_g^2\lambda_t^2\left(\mathbb{E}\|e_t^R\|^2 + L_g^2\mathbb{E}\|v_t - v_t^*\|^2\right)$$

$$+ 96(1 - \eta_{t+1}^R)^2\left(L_{g_{yy}}^2 r_v^2 + L_{f_y}^2\right)\left[\alpha_t^2\mathbb{E}\|\widetilde{h}_t^f\|^2 + 2\beta_t^2(\mathbb{E}\|e_t^g\|^2 + \mathbb{E}\|\nabla_y g(x_t, y_t)\|^2)\right]$$

*for all $t \in \{0, 1, ..., T-1\}$.*

As shown in Proposition 2, the LS gradient estimation error $e_{t+1}^R$ contains an iteratively improved error component $[(1 - \eta_{t+1}^R)^2(1 + 96L_g^4\lambda_t^2) + 4L_{g_{yy}}r_v^2\delta_\epsilon]\mathbb{E}\|e_t^R\|^2$ for the stepsize $\lambda_t$ and the approximation factor $\delta_\epsilon$ sufficiently small, a finite-difference approximation error $\mathcal{O}(\delta_\epsilon)$ as well as an approximation error $\mathcal{O}(\lambda_t^2\mathbb{E}\|v_t - v_t^*\|^2)$ for solving the LS problem. The next step is to upper-bound $\mathbb{E}\|v_t - v_t^*\|^2$.

**Proposition 3.** *Under the Assumption 1, 2, the iterates of the LS problem by Algorithm 1 satisfy*

$$\mathbb{E}\|v_{t+1} - v_{t+1}^*\|^2$$

$$\leq (1 + \gamma_t')\left(1 + \delta_t'\right)\left[\left(1 - 2\lambda_t\frac{(L_g + L_g^3)\mu_g}{\mu_g + L_g} + \lambda_t^2 L_g^2\right)\mathbb{E}\|v_t - v_t^*\|^2\right]$$

$$+ (1 + \gamma_t')\left(1 + \frac{1}{\delta_t'}\right)\lambda_t^2\mathbb{E}\|e_t^R\|^2$$

$$+ (1 + \frac{1}{\gamma_t'})\left(\frac{2L_{f_y}^2}{\mu_g^2} + \frac{2C_{f_y}^2 L_{g_{yy}}^2}{\mu_g^4}\right)\left[\alpha_t^2\mathbb{E}\|\widetilde{h}_t^f\|^2 + \beta_t^2\left(2\mathbb{E}\|e_t^g\|^2 + 2\mathbb{E}\|\nabla_y g(x_t, y_t)\|^2\right)\right].$$

*for all $t \in \{0, ..., T-1\}$ with some $\gamma'_t > 0$ and $\delta'_t > 0$.*

Based on the above important properties, we now provide the general convergence theorem for FdeHBO.

**Theorem 1.** *Suppose Assumptions 1, 2, 3 and Lemma 3 are satisfied. Choose $r_v \geq \frac{C_{f_y}}{\mu_g}$ and set*

$$\alpha_t = \frac{1}{(w+t)^{1/3}}, \quad \beta_t = c_\beta \alpha_t, \quad \lambda_t = c_\lambda \alpha_t, \quad \eta_t^f = c_{\eta_f} \alpha_t^2, \quad \eta_t^R = c_{\eta_R} \alpha_t^2, \quad \eta_t^g = c_{\eta_g} \alpha_t^2,$$

*and $\delta_\epsilon \leq \frac{\min\{c_{\eta_f}, c_{\eta_R}\}}{8(L_{g_{xy}} r_v^2 (w+T-1)^{2/3})}$, where the constants $w$, $c_\beta, c_\lambda, c_{\eta_f}, c_{\eta_R}$ and $c_{\eta_g}$ are defined in eq. (67) in the appendix. Then, the iterates generated by Algorithm 1 satisfy*

$$\mathbb{E}\|\nabla\Phi(x_a(T))\|^2 \leq \widetilde{\mathcal{O}}\left(\frac{\Phi(x_0) - \Phi^*}{T^{2/3}} + \frac{\|y_0 - y^*(x_0)\|^2}{T^{2/3}} + \frac{\|v_0 - v^*(x_0, y_0)\|^2}{T^{2/3}}\right.$$
$$\left. + \frac{1}{T^{2/3}} + \frac{\sigma_f^2}{T^{2/3}} + \frac{\sigma_g^2}{T^{2/3}} + \frac{\sigma_R^2}{T^{2/3}}\right).$$

**Corollary 1.** *Under the same setting of Theorem 1, FdeHBO requires $\widetilde{\mathcal{O}}(\epsilon^{-1.5})$ samples and gradient evaluations, respectively, to achieve an $\epsilon$-accurate stationary point.*

It can be seen from Corollary 1 that the proposed FdeHBO achieves an $\widetilde{\mathcal{O}}(\epsilon^{-1.5})$ sample complexity without any second-order derivative computation. As far as we know, this is the first Hessian/Jacobian-free stochastic bilevel optimizer with an $\widetilde{\mathcal{O}}(\epsilon^{-1.5})$ sample complexity.

### 3.3 Convergence and Complexity Analysis of FMBO

In this section, we analyze the convergence and complexity of the simplified FMBO method.

**Theorem 2.** *Suppose Assumptions 1, 2 and 3 are satisfied. Choose $r_v \geq \frac{C_{f_y}}{\mu_g}$ and set parameters*

$$\alpha_t = \frac{1}{(w+t)^{1/3}}, \quad \beta_t = c_\beta \alpha_t, \quad \lambda_t = c_\lambda \alpha_t,$$
$$\eta_t^f = c_{\eta_f} \alpha_t^2, \quad \eta_t^R = c_{\eta_R} \alpha_t^2, \quad \eta_t^g = c_{\eta_g} \alpha_t^2$$

*where $w$, $c_\beta, c_\lambda, c_{\eta_f}, c_{\eta_R}$ and $c_{\eta_g}$ are defined in eq. (33) in the appendix. The iterates generated by Algorithm 2 satisfy*

$$\mathbb{E}\|\nabla\Phi(x_a(T))\|^2 \leq \widetilde{\mathcal{O}}\left(\frac{\Phi(x_0) - \Phi^*}{T^{2/3}} + \frac{\|y_0 - y^*(x_0)\|^2}{T^{2/3}}\right.$$
$$\left. + \frac{\|v_0 - v^*(x_0, y_0)\|^2}{T^{2/3}} + \frac{\sigma_f^2}{T^{2/3}} + \frac{\sigma_g^2}{T^{2/3}} + \frac{\sigma_R^2}{T^{2/3}}\right).$$

Theorem 2 shows that the proposed fully single-loop FMBO achieves a convergence rate of $\frac{1}{T^{2/3}}$, which further yields the following complexity result.

**Corollary 2.** *Under the same setting of Theorem 2, FMBO requires totally $\widetilde{\mathcal{O}}(\epsilon^{-1.5})$ data samples, gradient and matrix-vector evaluations, respectively, to achieve an $\epsilon$-accurate stationary point.*

Corollary 2 shows that FMBO requires a total number $\widetilde{\mathcal{O}}(\epsilon^{-1.5})$ of data samples, which matches the best sample complexity in [34, 62, 28]. More importantly, each iteration of FMBO contains only one Hessian-vector computation due to the simple fully single-loop implementation, whereas other momentum-based approaches require $\mathcal{O}(\log\frac{1}{\epsilon})$ Hessian-vector computations in a nested manner per iteration. Also, note that FMBO is the first fully single-loop bilevel optimizer that achieves the $\widetilde{\mathcal{O}}(\epsilon^{-1.5})$ sample complexity.

## 4 Experiments

In this section, we test the performance of the proposed FdeHBO and FMBO on two applications: hyper-representation and data hyper-cleaning, respectively.

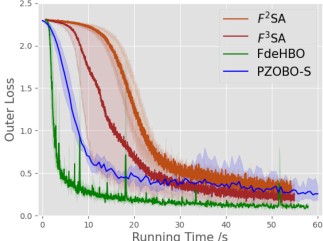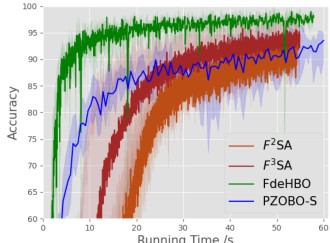

Figure 1: Comparison on hyper-representation with the LeNet neural network. Left plot: outer loss v.s. running time; right plot: accuracy v.s. running time.

## 4.1 Hyper-representation on MNIST Dataset

We now compare the performance of our Hessian/Jacobian-free FdeHBO with the relevant Hessian/Jacobian-free methods PZOBO-S [58], F$^2$SA [37] and F$^3$SA [37]. We perform the hyper-representation with the 7-layer LeNet network [38], which aims to solve the following bilevel problem.

$$\min_{\lambda} L_{\nu}(\lambda) := \frac{1}{|S_{\nu}|} \sum_{(x_i,y_i)\in S_{\nu}} L_{CE}(w^*(\lambda)f(\lambda; x_i), y_i)$$

$$s.t. \quad w^*(\lambda) = \arg\min_{w} L_{in}(\lambda, w), \quad L_{in}(\lambda, w) := \frac{1}{|S_{\tau}|} \sum_{(\tau,y_i)\in S_{\tau}} L_{CE}(wf(\lambda, x_i), y_i),$$

where $L_{CE}$ denotes the cross-entropy loss, $S_{\nu}$ and $S_{\tau}$ denote the training data and validation data, and $f(\lambda; x_i)$ denotes the features extracted from the data $x_i$. More details of the experimental setups are specified in Appendix A.1.

As shown in Figure 1, our FdeHBO converges much faster and more stably than PZOBO-S, F$^2$SA and F$^3$SA, while achieving a higher training accuracy. This is consistent with our theoretical results, and validates the momentum-based approaches in reducing the variance during the entire training.

## 4.2 Hyper-cleaning on MNIST Dataset

We compare the performance of our FMBO to various bilevel algorithms including AID-FP [21], reverse[17], SUSTAIN [34], MRBO and VRBO [62], BSA [19], stocBiO [30], FSLA [40] and SOBA [11], on a low-dimensional data hyper-cleaning problem with a linear classifier on MNIST dataset, which takes the following formulation.

$$\min_{\lambda} L_{\nu}(\lambda, w^*) = \frac{1}{|S_{\nu}|} \sum_{(x_i,y_i)\in S_{\nu}} L_{CE}((w^*)^T x_i, y_i)$$

$$s.t. \quad w^* = \arg\min_{w} L(\lambda, w) := \frac{1}{|S_{\tau}|} \sum_{(x_i,y_i)\in S_{\tau}} \sigma(\lambda_i) L_{CE}(w^T x_i, y_i) + C\|w\|^2, \quad (14)$$

where $L_{CE}$ denotes the cross-entropy loss, $S_{\nu}$ and $S_{\tau}$ denote the training data and validation data, whose sizes are set to 20000 and 5000, respectively, $\lambda = \{\lambda_i\}_{i\in S_{\tau}}$ and $C$ are the regularization parameters, and $\sigma(\cdot)$ is the sigmoid function. AmIGO [3] is not included in the figures because it performs similarly to stocBiO. The experimental details can be found in Appendix A.2.

As shown in Figure 2(a), FMBO, stocBiO and AID-FP converge much faster and more stable than other algorithms. Compared to stocBiO and AID-FP, FMBO achieves a lower training loss. This demonstrates the effectiveness of momentum-based variance reduction in finding more accurate iterates. It can be seen from Figure 2(b) that FMBO converges faster than existing fully single-loop FSLA and SOBA algorithms with a lower training loss.

## 5 Conclusion

In this paper, we propose a novel Hessian/Jacobian-free bilevel optimizer named FdeHBO. We show that FdeHBO achieves an $\mathcal{O}(\epsilon^{-1.5})$ sample complexity, which outperforms existing algorithms of the

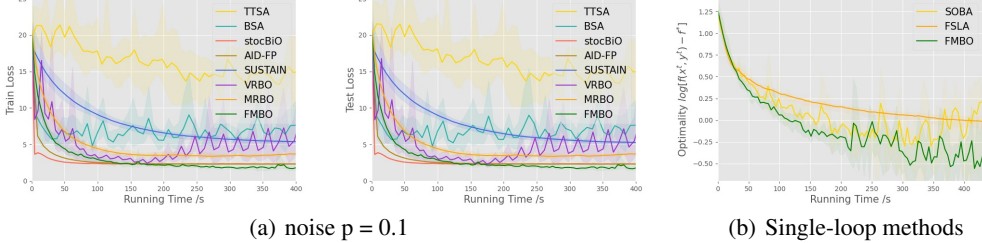

(a) noise p = 0.1          (b) Single-loop methods

Figure 2: (a) Comparison of different algorithms on data hyper-cleaning with noise $p = 0.1$. Left plot: test loss v.s. running time; right plot: train loss v.s. running time. (b) Comparison among different single-loop algorithms: training loss v.s. running time.

same type by a large margin. Our experiments validate the theoretical results and the effectiveness of the proposed algorithms. We anticipate that the developed analysis will shed light on developing provable Hessian/Jacobian-free bilevel optimization algorithms and the proposed algorithms may be applied to other applications such as fair machine learning.

## Acknowledgement

The work is supported in part by NSF under grants 2326592 and 2311274.

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
