# OpenReview forum: "Achieving $\mathcal{O}(\epsilon^{-1.5})$ Complexity in Hessian/Jacobian-free Stochastic Bilevel Optimization"
_NeurIPS.cc/2023/Conference — NeurIPS 2023 poster_

### Official Review · Reviewer_GD5q · 2023-06-22

**Soundness:** 2 fair
**Presentation:** 2 fair
**Contribution:** 2 fair
**Rating:** 5
**Confidence:** 4

**Summary:**

This paper proposed faster first-order stochastic algorithms for bilevel optimization, based on the idea that a Hessian-vector product can be implemented by finite difference of the gradient along the given direction.


**Strengths:**

The paper studies a very important problem and the provided complexity upper bound strictly improves previous works.

**Weaknesses:**

1. It is well-known that the Hessian-vector product can be implemented by the finite difference of the gradient along the given direction.
And the computation of a Hessian-vector product is known to be as easy as the computation of gradients. Therefore the results and algorithms seem not to be novel compared to the existing methods (AID/ITD) that use Hessian-vector products as oracles.

2. The paper uses three different names for the same baseline, including F${}^2$SA, F2SA, $F{}^2$SA  (Line 442, Table 1, Figure 1). Should use a unified name.





**Questions:**

I have a question about the claim of "near-optimal''.
Assumption 3 supposes that "Assumptions 1 and 2 hold for $f(x,y;\xi)$ and $g(x, y;\zeta)$ for $\forall \xi$ and $\forall \zeta$."
But according to "Lower Bounds for Non-Convex Stochastic  Optimization"(https://arxiv.org/pdf/1912.02365.pdf), the lower bound function is constructed under the mean-square-smooth Assumption. Therefore it seems that the mean-square-smoothness should be imposed for $f$, which is slightly stronger than the smoothness of $f(x,y;\xi)$ for $\forall \xi$?

I think it may be a fixable bug. But the paper may need some modifications.

Please let me know if I am wrong. I will adjust my ratings based on the rebuttal.

---

> ### Author Rebuttal · Authors · 2023-08-09
>
> We thank the reviewer GD5q for the time and valuable feedback!
>
> **Q1: The paper uses three different names for the same baseline, including FSA, F2SA, FSA (Line 442, Table 1, Figure 1). Should use a unified name.**
>
> **A:** Many thanks! We will use a unified name in the revision.
>
> **Q2: I have a question about the claim of "near-optimal''. Assumption 3 supposes that "Assumptions 1 and 2 hold for $f(x,y; \xi)$ and $g(x,y; \zeta)$ for $\forall \xi$ and $\forall \zeta$. But the lower bound function is constructed under the mean-square-smooth assumption. Therefore it seems that the mean-square-smoothness should be imposed for $f$, which is slightly stronger than the smoothness of $f(x,y;\xi)$ for $\forall \xi$?**
>
> **A:** Good question! Note that if our assumption on the smoothness of $f(x,y;\xi)$ for $\forall \xi$ holds, then the mean-square-smooth assumption (eq. (4) in [2] pointed out by the reviewer) must hold. This means that the problem class considered in our paper is a subset of that in [2]. Thus, if there exists a case in our problem class that achieves a lower bound smaller than $\Omega(\epsilon^{-1.5})$, then since this case also belongs to that of [2], the lower bound in [2] should also be smaller than $\Omega(\epsilon^{-1.5})$. This contradicts the result in [2]. Thus, by contradiction, we can show that the lower bound under our assumptions is larger than or equal to $\Omega(\epsilon^{-1.5})$. Since our proved upper bound of $O(\epsilon^{-1.5})$ matches this $\Omega(\epsilon^{-1.5})$ lower bound, we can claim that our algorithm is “near-optimal”.
>
> [2] Arjevani, Yossi, et al. "Lower bounds for non-convex stochastic optimization." Mathematical Programming 199.1-2 (2023): 165-214.

---

> > ### Comment · Reviewer_GD5q · 2023-08-11
> > **Response to the Rebuttal**
> >
> > ## Novelty
> >
> > This paper is based on $ \nabla^2 f(x) v \approx (\nabla f(x+\delta v) - \nabla f(x)) / \delta$. It is a well-known fact. See for example Page 16-1 in this lecture (https://drive.google.com/file/d/1EzIXS42TY47KMl9T99OqXUYzKsOYiYHy/view). The result of this paper is already known to me before reading this paper.
> >
> > ## Lower Bound
> >
> > What the authors say in the rebuttal does not make any sense. I think the authors have some misunderstanding of the meaning of lower bound.
> >
> > I definitely think this paper should not be accepted, and I decide to decrease my score to 3.

---

> > > ### Author Response · Authors · 2023-08-11
> > > **Thanks for the quick response**
> > >
> > > We thank you for the very quick response and we realized we made an improper claim here. Indeed, we cannot claim that the lower bound in our case is larger than $\epsilon^{-1.5}$, by the following observations:
> > >
> > > The meaning of lower bound (we need the following concepts):
> > >
> > > Problem class $C$: a class of objective functions that satisfy the assumptions
> > >
> > > Algorithm class $A$: a class of gradient-based algorithms of interest
> > >
> > > The lower bound means that there exists an example in $C$ such that for any algorithm in $A$, the number of samples is at least $\Omega(\epsilon^{-1.5})$.
> > >
> > > Then, in our case:
> > >
> > > Let $C_1$ be the problem class that satisfy the assumptions based on mean-square-smooth assumption. Let $C_2$ be the problem class that satisfy our assumptions. We consider the same algorithm class $A$.
> > >
> > > Then, suppose we get a lower bound $L_2$, which means that there exists an example $E_2$ in $C_2$ such that for any algorithm in $A$, the number of samples is at least $L_2$. Since this example $E_2$ also belongs to $C_1$, the lower bound $L_1$ for $C_1$ (solved by $A$) should be larger than $L_2$.
> > >
> > > Thus, what we can claim is that our complexity matches the optimal sample complexity for single-level problem satisfying the mean-square-smooth assumption. We will revise our statements in our paper. However, achieving the $\Theta(\epsilon^{-1.5})$ upper bound in our Hessian-free bilevel optimization (under the same assumptions used in all previous bilevel works)  is already a good contribution itself, so I wish the reviewer can also recognize this.

---

> > > ### Author Response · Authors · 2023-08-17
> > > **Thanks for raising this concern for us**
> > >
> > > Dear Reviewer GD5q,
> > >
> > > We thank you again for raising the issue with "near-optimal", and we are also sorry about providing you an improper explanation in our first response. After we carefully check the lower bounds under our assumptions (based on our second response to you), we indeed find that our claim on "near-optimal" is improper. However, since the issue with "near-optimal" only appears in the statements rather than in proofs, we believe it is fixable by revising the statements carefully (e.g., directly referring to $O(\epsilon^{-1.5})$ instead of claiming it to be "near-optimal").
> > >
> > > For the contributions, although stochastic Hessian-free bilevel optimization has been studied recently (e.g., F$^2$SA and F$^3$SA with momentum, or some other value function based methods), the best upper bound is $O(\epsilon^{-2.5})$ under the same assumptions as ours. Thus, achieving a $O(\epsilon^{-1.5})$ complexity itself should be a big complement to existing studies on Hessian-free bilevel optimization.
> > >
> > > In terms of the novelty, although the finite-difference Hessian-vector estimation is not new, applying it in stochastic Hessian-free bilevel optimization is non-trivial. For example, we introduce an auxiliary projection to guarantee the boundedness of this estimation error, and we also need to carefully deal with the biased momentum-based gradient approximations. These analyses are not straightfoward  particularly in achieving the $O(\epsilon^{-1.5})$ complexity.
> > >
> > > We thank the reviewer again for the time and for raising this issue for us! We will carefully revise the claims and statements in the revision.
> > >
> > > Best,
> > > Authors

---

> > > > ### Comment · Reviewer_GD5q · 2023-08-18
> > > > **Further Discusssions**
> > > >
> > > > I appreciate that the authors can realize their mistakes. The rating I gave earlier was based on not wanting to see an article published with a fatal error in even the title. I sincerely hope that the author can carefully correct it.
> > > >
> > > > Given that the research topic of the article is very important, and people in this field have not realized the result given by this article before, I think the article still has its theoretical contributions.
> > > >
> > > > The finite-difference Hessian-vector estimation has been studied a lot when studying the problems of escaping from saddle points. I hope the author can take a look at related works and cite them when necessary.
> > > > For example, some similar techniques in this article seemed to have appeared in [1].
> > > > Maybe some of the conclusions in [1] can be cited directly. Then the authors may avoid making some mistakes in their original proof (as Reviewer ySP3 has pointed out).
> > > >
> > > > It is worth noting that previous works usually regarded the use of finite-difference approximations to the Hessian is often cited as a **disadvantage** rather than an advantage, see for example [2] says:
> > > > "When the function has a Hessian Lipschitz property such an oracle can be approximated by differentiating the gradients at two very close points (although this may suffer from numerical issues,
> > > > thus is seldom used in practice)."
> > > >
> > > > In my experience, finite-difference approximations are often not practical.
> > > > In fact, this is because the oracle model does not consider the problem of rounding errors. If taking the rounding errors into account, then the distance of finite differences cannot be arbitrarily small. A similar discussion appears in Chapter 9 of this book [3].
> > > >
> > > > The aim of many Hessian-vector-free methods is therefore to implement first-order algorithms that do not rely on finite differences, such as [2]. I would call such an algorithm  (such as F${}^2$SA) a true first-order algorithm. Hence I think the contribution of this paper is more on the theoretical side than on the practical side because the proposed algorithm is not a true first-order algorithm from my point of view.
> > > > I think more important future work is to realize a **true** near-optimal first-order algorithm.
> > > > In fact, recent research [4] seems to have addressed this issue in the deterministic case.
> > > >
> > > > [1] Neon2: Finding local minima via first-order oracles. Z Allen-Zhu, Y Li . In NeurIPS, 2018.
> > > >
> > > > [2] How to escape saddle points efficiently. Jin, Chi, et al. In ICML, 2017.
> > > >
> > > > [3] Numerical optimization. Second Edition. N Jorge, JW Stephen.
> > > >
> > > > [4] Near-Optimal Fully First-Order Algorithms for Finding Stationary Points in Bilevel Optimization. arXiv preprint, 2023.

---

> > > > > ### Author Response · Authors · 2023-08-18
> > > > > **Thanks for the feedback**
> > > > >
> > > > > Dear Reviewer GD5q,
> > > > >
> > > > > We thank you so much for the further feedback and the valuable suggestions! We will carefully correct improper claims in the revision, and also provide a detailed discussions on the related works such as [1] that use the technique of finite-difference Hessian-vector estimation. The discussions on the possible rounding errors will also be discussed.
> > > > >
> > > > > We are also happy to see that the reviewer recognizes the theoretical contributions of our work. Achieving $O(\epsilon^{-1.5})$ complexity **without** finite-difference Hessian-vector estimation seems to be still open at the current stage, but we would like to take efforts along this direction for future study.
> > > > >
> > > > > We thank the reviewer again for the valuable comments!
> > > > >
> > > > > Best,
> > > > > Authors

---

> > > > > > ### Comment · Reviewer_GD5q · 2023-08-18
> > > > > > **Thanks again**
> > > > > >
> > > > > > I have updated my rating.

---

> > > > > > > ### Author Response · Authors · 2023-08-18
> > > > > > > **Thanks!**
> > > > > > >
> > > > > > > Dear Reviewer GD5q,
> > > > > > >
> > > > > > > We sincerely thank you for the updates! We will definitely take your suggestions into our revision!
> > > > > > >
> > > > > > > Best,
> > > > > > > Authors

---

### Official Review · Reviewer_ySP3 · 2023-07-02

**Soundness:** 3 good
**Presentation:** 3 good
**Contribution:** 2 fair
**Rating:** 5
**Confidence:** 4

**Summary:**

This paper provides a novel Hessian-free bilevel optimization algorithm called FdeHBO, which improves upon previous state of the art F3SA with a better iteration complexity. The author provide convergence analysis and empirical results to show the strength of the proposed method. However, I have some concerns regarding the proof, which is detailed in the weaknesses part.

**Strengths:**

The algorithm is novel, as far as I am concerned, and the improved convergence rate seems promising. The paper is well organized and easy to follow. The experiment seems convincing.

**Weaknesses:**

The proof of Lemma 5 is incorrect. In the proof,
the authors attempt to use the mean value theorem for a function $\mathbb{R}\to \mathbb{R}^n$, but this is in general wrong. The estimation error of the finite difference of in the lemma is a key step, without which the proof of the main theorem is incorrect.

In general, a zeroth order optimization algorithm should have a dimension dependency, but this paper magically removes this. I believe the one of the reason is the mistake in Lemma 5. I will be glad to change my evaluation if I am wrong.

**Questions:**

See weakness part above.

**Limitations:**

I do not see the discussion of limitations, but I think this is not a major problem. I do not see potential negative societal impact as this paper mainly focuses on the theoretical side.

---

> ### Author Rebuttal · Authors · 2023-08-09
>
> We thank the reviewer ySP3 for the time and valuable feedback!
>
> **Q1: The proof of Lemma 5 is incorrect. In the proof, the authors attempt to use the mean value theorem for a function $R \rightarrow R^n$, but this is in general wrong. The estimation error of the finite difference in the lemma is a key step, without which the proof of the main theorem is incorrect.**
>
> **A:** We thank the reviewer for pointing this out for us! For rigorous derivations, we provide a very detailed proof below for Lemma 5, which includes a generalized mean value theorem for vector-valued function, as well as a refined upper bound on the finite-difference estimation error. It can be seen that our characterizations on the estimation error $\\|e_t^J\\|^2$ are still bounded by a constant $C_J$ multiplied with a tunable sufficiently small parameter $\delta_\epsilon$ (similarly for the error  $\\|e_t^H\\|^2$). Thus, the remaining steps will not be affected. We will add such details in the revision.
>
> **Q2: In general, a zeroth order optimization algorithm should have a dimension dependency, but this paper magically removes this. I believe one of the reasons is the mistake in Lemma 5.**
>
> **A:** We would like to explain this dimension-free result from three perspectives.
>
> First, intuitively speaking, a typical zeroth-order method has such dimension dependence because it uses a one-dimension function value to approximate all $n$ coordinates of a $n$-dimension gradient vector. However, in our case, the finite-difference approach uses the $n$-dimension gradient information to approximate the $n$-dimension Hessian-vector product rather than the $n^2$-dimension Hessian matrix itself. More technically, one key property here is that the smoothing direction vector in our estimator is exactly the same as the vector of the Hessian-vector product. This is why the dimension dependence disappears in our results.
>
> Second, rigorously speaking, we provide detailed proof below. It can be seen that the upper bound does not contain any dimension dependence.
>
> Third, we also note that similar results have been derived in previous studies, e.g., see the inequality right after eq.(140) in [1], where it can be seen that the finite-difference estimation error bound does not contain any dimension dependence either.
>
> [1] Fallah, Alireza, Aryan Mokhtari, and Asuman Ozdaglar. "On the convergence theory of gradient-based model-agnostic meta-learning algorithms."
>
> ---
>
> [**The mean-value theorem for vector-valued function**]
> Let $S \subseteq \mathbb{R}^n$ be open and let $f: S \rightarrow \mathbb{R}^m$ be differentiable on all of $S$. Let $x,y \in S$ be such that the line segment connecting these two points is contained in $S$, i.e., $L(x,y) = \\{ (1-t)x+ty: t\in [0,1]\\} \subseteq S$. Then for every $\mathbf{a} \in \mathbb{R}^m$, there exists a point $z \in L(x,y)$ such that $\mathbf{a} \cdot [f(y)-f(x)] = \ \mathbf{a} \cdot [f'(z)(y-x)]$.
>
> Proof: Let $\mathbf{a} \in \mathbb{R}^m$ and define a new function $F:[0,1] \rightarrow \mathbb{R}$ for all $t \in [0,1]$ by: $F(t) = \mathbf{a} \cdot f(x+t(y-x))$. Since $f$ is differentiable on $S$, we have
> that $F$ must be continuous on $[0,1]$. Furthermore, $F$ is differentiable on $(0,1)$ by the chain rule: $F'(t) = \mathbf{a} \cdot f'(x+t(y-x))(y-x)$. So by the Mean Value Theorem for single-variable real-valued functions, there exists a number $h \in (0,1)$ for which:
> $$F(1) - F(0) = F'(h)(1-0),$$
> where the left-hand side is
> $$F(1) - F(0) = \mathbf{a} \cdot f(y) - \mathbf{a} \cdot f(x)$$
> and the right-hand side is
> $$F'(h)(1-0) = \mathbf{a} \cdot f'(x+h(y-x))(y-x).$$
> Set $z = x+h(y-x)$, then $z \in L(x,y)$ and we have from the equality that
> $$\mathbf{a} \cdot [f(y) - f(x)] = \mathbf{a} \cdot f'(z)(y-x).$$
>
> [**Rewrite Lemma 5**]
> Under Assumption 2, for any $\xi_t$ and $\psi_t$, define $$e_t^J:= \widetilde{J}(x_t, y_t, v_t, \delta_{\epsilon};\xi_t) - \nabla^2_{xy}g(x_t,y_t;\xi_t)v_t, \quad e_t^H:= \widetilde{H}(x_t, y_t, v_t, \delta_{\epsilon};\psi_t) - \nabla^2_{yy}g(x_t,y_t;\psi_t)v_t,$$
> then we have the bounds of $e_t^J$ and $e_t^H$ as
> $$\\|e_t^J\\|^2 \leq C_J\delta_{\epsilon}, \quad \\|e_t^H\\|^2 \leq C_H\delta_{\epsilon}, $$
> where $C_J := 4L_gL_{g_{xy}}r_v^3$, $C_H :=4L_gL_{g_{yy}}r_v^3$.
>
> Proof: Here, in our case, we have
> $$\mathbf{a} \cdot \bigg[\frac{\nabla_y g(x_t,y_t+\delta_{\epsilon}v_t;\zeta_t) - \nabla_y g(x_t,y_t-\delta_{\epsilon}v_t;\zeta_t)}{2\delta_{\epsilon}} - \nabla_{yy}^2 g(x_t,y_t;\zeta_t)v\bigg] \ = \mathbf{a} \cdot \Big[\nabla_{yy}^2 g(x_t,y_t+\delta_{\epsilon}'v_t;\zeta_t) - \nabla_{yy}^2 g(x_t,y_t;\zeta_t)\Big] $$
> Let $\mathbf{a} = \bigg[\Big(\nabla_y g(x_t,y_t+\delta_{\epsilon}v_t;\zeta_t) - \nabla_y g(x_t,y_t-\delta_{\epsilon}v_t;\zeta_t)\Big)/(2\delta_{\epsilon}) - \nabla_{yy}^2 g(x_t,y_t;\zeta_t)v_t\bigg]^T$, then we have
>
> \begin{align}
> \\|e_t^H\\|^2 &= \bigg[\frac{\nabla_y g(x_t,y_t+\delta_{\epsilon}v_t;\zeta_t) - \nabla_y g(x_t,y_t-\delta_{\epsilon}v_t;\zeta_t)}{2\delta_{\epsilon}} - \nabla_{yy}^2 g(x_t,y_t;\zeta_t)v_t\bigg]^T  \cdot \Big[\nabla_{yy}^2 g(x_t,y_t+\delta_{\epsilon}'v_t;\zeta_t) - \nabla_{yy}^2 g(x_t,y_t;\zeta_t)\Big]
> \end{align}
> \begin{align}
> \leq \bigg\\|\frac{\nabla_y g(x_t,y_t+\delta_{\epsilon}v_t;\zeta_t) - \nabla_y g(x_t,y_t-\delta_{\epsilon}v_t;\zeta_t)}{2\delta_{\epsilon}} \bigg\\| \cdot \Big\\|\nabla_{yy}^2 g(x_t,y_t+\delta_{\epsilon}'v_t;\zeta_t) - \nabla_{yy}^2 g(x_t,y_t;\zeta_t)\Big\\|
> \end{align}
> \begin{align}
>  \+ \big\\|\nabla_{yy}^2 g(x_t,y_t;\zeta_t)v_t\big\\| \cdot \Big\\|\nabla_{yy}^2 g(x_t,y_t+\delta_{\epsilon}'v_t;\zeta_t) - \nabla_{yy}^2 g(x_t,y_t;\zeta_t)\Big\\|
> \end{align}
> \begin{align}
> \overset{(a)}{\leq} 4L_gL_{g_{yy}}\\|v_t\\|^3\delta_{\epsilon}' \leq 4L_gL_{g_{yy}}r_v^3\delta_{\epsilon} ,\nonumber
> \end{align}
> where (a) uses the Lipschitz continuity of $\nabla_y g(x,\cdot)$ and $\nabla^2_{yy} g(x,\cdot)$ in Assumption 2.
> Similarly, we have $\\|e_t^H\\|^2 \leq 4L_gL_{g_{xy}}r_v^3\delta_{\epsilon}$.

---

> > ### Comment · Reviewer_ySP3 · 2023-08-17
> >
> > Thanks for the clarification. I will increase my score.
> >
> > I also read the questions raised by other reviewers. I agree with Reviewer GD5q that "near optimal" is an overclaim, and recommend the authors to revise it carefully.

---

> > > ### Author Response · Authors · 2023-08-17
> > > **Thanks for your reply!**
> > >
> > > Dear Reviewer ySP3,
> > >
> > > We thank you so much for the feedback and for increasing the score! Yes, we agree that the "near-optimal" is an overclaim after we carefully check the lower bounds under our assumptions. We are sorry about this. We will carefully revise the statements in our paper (e.g., directly referring to $O(\epsilon^{3/2})$ instead of claiming it as "near-optimal").
> > >
> > > We thank you again for your time!
> > >
> > > Best,
> > > Authors

---

### Official Review · Reviewer_HtEu · 2023-07-06

**Soundness:** 3 good
**Presentation:** 3 good
**Contribution:** 3 good
**Rating:** 7
**Confidence:** 3

**Summary:**

This work focuses on addressing a stochastic bilevel optimization problem characterized by a strongly convex lower-level objective function and a nonconvex upper-level objective. The authors specifically investigate a Hessian-free (fully first-order) approach to bilevel optimization, which avoids the need for Hessian or Jacobian-vector computations. They introduce a new method called FdeHBO, which adopts a single-loop structure and incorporates momentum-based updates for the upper-level problem, linear system, and lower-level problem.
FdeHBO achieves a sample and gradient complexity of $O(\epsilon^{-1.5})$, which is the first result for nonconvex-strongly-convex Hessian-free bilevel optimization. Additionally, the authors propose a simplified and nearly optimal variant called FMBO, which is specifically designed for scenarios with small-dimensional problems. To validate the effectiveness of their proposed methods, the authors conduct several experiments.

**Strengths:**

1. The paper is overall easy to follow. The topic of Hessian-free bilevel optimization, particularly in the nonconvex-strongly-convex setting, has not been extensively explored. Therefore, the investigation into the development of simple and nearly optimal stochastic Hessian-free bilevel algorithms is both interesting and significant.
2. To the best of my knowledge, the proposed FdeHBO algorithm achieves near-optimal sample/gradient complexity for the first time. The utilization of the projection-aided finite-difference estimator in bilevel optimization is interesting. The experimental results depicted in Figure 1 provide substantial evidence of FdeHBO's superior performance compared to other approaches of the same kind.
3. From a technical standpoint, addressing issues such as momentum-based errors, biased estimation caused by the finite-difference estimator, and auxiliary projection errors requires nontrivial treatments, further highlighting the sophistication of this work.

**Weaknesses:**

1. The paper's projection technique is limited to projecting onto a ball. Is it possible that  this approach can be generalized to encompass projections onto more general convex and closed sets?

2. While this work primarily focuses on the momentum-based estimator, it appears that the approach can be simplified to a more general stochastic gradient descent (SGD)-type method by setting $\eta_t^f = \eta_t^g = \eta_t^R = 0$. Does the analysis presented in the paper adequately cover this specific case? It is worth considering the importance of simpler SGD updates with reduced tuning requirements, as they often hold practical significance.

3. The authors should provide further elaboration on why the auxiliary projection does not impact the final convergence guarantee. Although I have not thoroughly examined all the proofs, a clearer explanation on this aspect would be beneficial.


**Questions:**

See Weakness.

$\nabla_y^2 g$ in Line 126 should be corrected to $\nabla_{yy}^2 g$?

**Limitations:**

yes

---

> ### Author Rebuttal · Authors · 2023-08-09
>
> We thank the reviewer HtEu for the time and valuable feedback!
>
> **Q1: The paper's projection technique is limited to projecting onto a ball. Is it possible that this approach can be generalized to encompass projections onto more general convex and closed sets?**
>
> **A:** Great question! The answer is yes. Our proof only requires this projection set to be convex and contain the optimal solution $v_t^*$. Since $v_t^*$ is bounded by $r_v$ (see Lemma 1), the proof still holds if we choose the convex set large enough. We will comment on this in the revision.
>
> **Q2: While this work primarily focuses on the momentum-based estimator, it appears that the approach can be simplified to a more general stochastic gradient descent (SGD)-type method by setting $\eta_t^f = \eta_t^g = \eta_t^R = 0$. Does the analysis presented in the paper adequately cover this specific case? It is worth considering the importance of simpler SGD updates with reduced tuning requirements, as they often hold practical significance.**
>
> **A:** Good question! As the reviewer noted, our framework reduces to SGD-type updates if we set $\eta_t^f = \eta_t^g = \eta_t^R = 0$, but the analysis may require proper adjustments in learning rate selection and variance characterization of gradient estimations. We would like to leave this for future study.
>
> **Q3: The authors should provide further elaboration on why the auxiliary projection does not impact the final convergence guarantee.**
>
> **A:** The projection appeals in two key steps in bounding the iterative gaps $E\\|v_{t+1} - v_t\\|^2$ in (25) and $E\\|v_{t+1} - v_t^*\\|^2$ in (28). In specific, by using the non-expansive property of projection that $\\|P_B(x)-P_B(y)\\|\leq \\|x-y\\|$ for a projection $P$ on a set $B$, we bound these two gaps by $\mathbb{E}\\|v_{t+1} - v_t\\|^2 \leq \mathbb{E}\\|w_{t+1} - v_t\\|^2 = \mathbb{E}\\|\lambda_t h_t^R\\|^2$ and
> $\mathbb{E}\\|v_{t+1} - v_t^*\\|^2 \leq \mathbb{E}\\|w_{t+1} - v_t^*\\|^2 = \mathbb{E}\\|v_t - \lambda_t h_t^R - v_t^*\\|^2$. The remaining steps are free from projection and the convergence will not be affected. We will clarify this in the revision.
>
> **Q4: $\nabla_y^2 g$ in line 126 should be corrected to $\nabla_{yy}^2 g$?**
>
> **A:** Yes, we will correct it in the revision.

---

> > ### Comment · Reviewer_HtEu · 2023-08-16
> >
> > Thank you for the clarification.

---

### Official Review · Reviewer_Xzpd · 2023-07-24

**Soundness:** 3 good
**Presentation:** 3 good
**Contribution:** 3 good
**Rating:** 6
**Confidence:** 2

**Summary:**

The paper studies stochastic bilevel optimization, where the first level objective function depends on the optimizer of the second level objective function. The paper provides an optimization procedure involving gradient computations and projections. The paper provides rates of convergence for the method. Since convexity is not assumed, convergence is expressed in terms of the gradient becoming small (not in terms of converging to a global optimizer). The benefit of the algorithm is demonstrated on various data sets.

**Strengths:**

The technical contribution of the paper appears to be impressive. The algorithm suggested is involved. Its novelties and benefits compared to the state of the art are discussed extensively. The convergence proofs appear to be involved and innovative. The numerical performances are very good, especially when compared to the state of the art.

**Weaknesses:**

The content is very technical, and it is not very easy to follow, especially given the space constraint in the main text.
In the appendix, the description of the experiments is also difficult to follow. There there are no space constraints, so I think the description could have been more pedagogical there.

**Questions:**

Algorithm 1: Hessain -> Hessian.

Conclusion, why is it written that the sample complexity is O(epsilon^-2), while Corollary 1 writes O(epsilon^-1.5)?

Appendix A.1. What are lambda and w? What are the dimensions?



**Limitations:**

The authors adequately discuss limitations.
I think there is not concern about societal impact.

---

> ### Author Rebuttal · Authors · 2023-08-09
>
> We thank reviewer Xzpd for the time and valuable feedback!
>
> **Q1: Suggestion on presentations.**
>
> **A:** We will definitely follow your suggestions to improve the presentation.
>
> **Q2: Conclusion, why is it written that the sample complexity is** $O(\epsilon^{-2})$**, while Corollary 1 writes** $O(\epsilon^{-1.5})$**?**
>
> **A:** Sorry about this typo. The sample complexity should be $O(\epsilon^{-1.5})$ in the conclusion. We will revise it.
>
> **Q3: In Appendix A.1, what are $\lambda$ and $w$? What are the dimensions?**
>
> **A:** We perform the hyper-representation with the 7-layer LeNet network to solve the bilevel problem. We take the last two layers as the lower-level parameters $w$, and all remaining layers as the upper-level parameters $\lambda$. In our experiments, the dimension of $w$ is 850, and the dimension of  $\lambda$ is 60856.

---

> > ### Comment · Reviewer_Xzpd · 2023-08-16
> > **Acknowledgement of rebuttal**
> >
> > Dear authors,
> > Thank your for the rebuttal.

---

### Comment · Area_Chair_qM8V · 2023-08-13

Thanks to all reviewers and authors for their work on this submission.

As the discussion period starts, I want to make sure that reviewers have read the author's response.

This can be done either by communicating with authors, or in private conversation within the reviewing team.

---

### Decision · Program_Chairs · 2023-09-21

**Decision:**

Accept (poster)

**Comment:**

This paper proposes a bilevel optimization algorithm that do not use the Hessian of the outer objective and that it has the best complexity known complexity. It was overall well received by the reviewers, and the rebuttal allowed to fix some important concerns.

It is **absolutely crucial** that the camera-ready acknowledges the fact that FdeHBO is not proved to be near-optimal by changing the title and the core of the text. It is also important that the authors discusses the fact if being Hessian-free is a *good* property. Indeed the computation of an HVP is marginally higher than a JVP. The paper does not really discuss this aspect, either from a theoretical aspect or from an experimental point of view.